# Synergistic Activity of Ketoconazole and Miconazole with Prochloraz in Inducing Oxidative Stress, GSH Depletion, Mitochondrial Dysfunction, and Apoptosis in Mouse Sertoli TM4 Cells

**DOI:** 10.3390/ijms23105429

**Published:** 2022-05-12

**Authors:** Sabrina Petricca, Giuseppe Celenza, Carla Luzi, Benedetta Cinque, Anna Rita Lizzi, Nicola Franceschini, Claudio Festuccia, Roberto Iorio

**Affiliations:** 1Department of Biotechnological and Applied Clinical Sciences, University of L’Aquila, Via Vetoio, 67100 L’Aquila, Italy; sabrina.petricca@univaq.it (S.P.); giuseppe.celenza@univaq.it (G.C.); carla.luzi@univaq.it (C.L.); annarita.lizzi@univaq.it (A.R.L.); nicola.franceschini@univaq.it (N.F.); claudio.festuccia@univaq.it (C.F.); 2Department of Life, Health and Environmental Sciences, University of L’Aquila, Via Vetoio, 67100 L’Aquila, Italy; benedetta.cinque@univaq.it

**Keywords:** fluconazole, prochloraz, miconazole and ketoconazole, mouse Sertoli TM4 cells, synergistic effects, oxidative stress, ROS generation and apoptosis, mitochondrial activity, SOD, enzymatic activity, GSH homeostasis

## Abstract

Triazole and imidazole fungicides represent an emerging class of pollutants with endocrine-disrupting properties. Concerning mammalian reproduction, a possible causative role of antifungal compounds in inducing toxicity has been reported, although currently, there is little evidence about potential cooperative toxic effects. Toxicant-induced oxidative stress (OS) may be an important mechanism potentially involved in male reproductive dysfunction. Thus, to clarify the molecular mechanism underlying the effects of azoles on male reproduction, the individual and combined potential of fluconazole (FCZ), prochloraz (PCZ), miconazole (MCZ), and ketoconazole (KCZ) in triggering in vitro toxicity, redox status alterations, and OS in mouse TM4 Sertoli cells (SCs) was investigated. In the present study, we demonstrate that KCZ and MCZ, alone or in synergistic combination with PCZ, strongly impair SC functions, and this event is, at least in part, ascribed to OS. In particular, azoles-induced cytotoxicity is associated with growth inhibitory effects, G0/G1 cell cycle arrest, mitochondrial dysfunction, reactive oxygen species (ROS) generation, imbalance of the superoxide dismutase (SOD) specific activity, glutathione (GSH) depletion, and apoptosis. N-acetylcysteine (NAC) inhibits ROS accumulation and rescues SCs from azole-induced apoptosis. PCZ alone exhibits only cytostatic and pro-oxidant properties, while FCZ, either individually or in combination, shows no cytotoxic effects up to 320 µM.

## 1. Introduction

Infertility is a global health issue affecting at least 50 million couples of reproductive age, and 12% of all men [1,2]. According to the “Global Burden of Disease Study”, between 1990 and 2017, the age-standardised prevalence of infertility has increased annually by 0.291% in men [3]. Several factors can contribute to male infertility, including hormonal imbalances, genetic factors, sexually transmitted diseases, environment, and lifestyle [4]. Concerning the idiopathic risk factors, environmental or occupational exposure to toxic chemicals (e.g., pesticides) represents a potential risk aspect for male infertility [5,6]. Indeed, environmental pollutants have been shown to have endocrine toxicity and hormone-like activity, resulting in changes in human male reproductive function and fertility alterations [7,8]. In this regard, multigenerational and transgenerational effects, as well as adverse reproductive outcomes on spermatogenesis and/or semen quality at both genetic and molecular levels, have been reported [9,10]. Several environmental toxicants can cause oxidative stress (OS), developmental and reproductive abnormalities, and infertility [11,12]. Therefore, OS is now considered as the main cause of male infertility [13]. In particular, in 30–80% of infertile men, sperm damage by reactive oxygen species (ROS) represents the key reason for infertility [14]. 

Sertoli cells (SCs) have emerged as one of the principal targets of environmental toxicants in male reproductive dysfunctions [15,16]. They play an essential role in spermatogenesis by providing support and nutrition for germ cells in the testis. They are also crucial in testicular function for expressing the androgen-binding protein (ABP) and the follicle-stimulating hormone receptor (FSHR), as well as in the formation of the blood-testis barrier (BTB) [17]. In performing this task, SCs also regulate and maintain the appropriate cellular redox status of sperm cells. During spermatogenesis, cell metabolism is highly activated by sex steroid hormones, indicating that germ cells themselves generate high levels of ROS. Moreover, they have limited antioxidant defence mechanisms and a restricted capacity to sense and repair DNA damage. Therefore, spermatozoa are strongly susceptible to OS and oxidative DNA damage [18]. SCs have evolved many mechanisms to suppress OS and minimise damage by ROS. In this sense, their antioxidant and redox systems support and cooperate (*helper function*) with spermatogenic cells, supplying them with glutathione (GSH) and superoxide dismutase (SOD), thus cooperating to compensate for their declining potency. Accordingly, xenobiotics-induced SC damage is critical in inducing male infertility.

Recent decades have seen the emergence of azole compounds as a new class of pesticides and active ingredients in therapeutic and personal care products [19]. Triazole and imidazole fungicides have been detected in surface water and sediment, soil environments, and agricultural products due to their physicochemical properties, including low volatility and biodegradability and high chemical stability [19,20]. The fungicidal activity of azoles results from their ability to inhibit lanosterol 14α-demethylase (CYP51), thereby affecting ergosterol biosynthesis, an important constituent of the fungal cell membrane [21]. Unfortunately, azoles can also interact with a wide range of mammalian cytochrome P450 enzymes (CYPs), including key enzymes involved in steroidogenesis (e.g., CYP19 aromatase). Consequently, individual or additive and synergistic anti-androgenic and –estrogenic activities, in addition to effects on the reproductive development of different azole mixtures, have been reported [22,23,24,25,26,27]. Interestingly, besides the interference with the steroid biosynthetic pathway, the imidazole and triazole derivatives have been shown to induce OS in different cellular models [28,29,30,31], indicating that free radicals might be responsible for azole-induced toxicity. In particular, a relationship between miconazole (MCZ)-, prochloraz (PCZ)-, ketoconazole (KCZ)-, and fluconazole (FCZ)-induced reactive oxygen species (ROS) generation, lipid peroxidation, alterations of glutathione content and antioxidant enzyme activities, as well as mitochondrial toxicity and cell death have been identified [32,33,34,35,36,37,38,39,40,41,42,43]. Concerning male reproduction, a joint activity of tebuconazole and econazole in inducing mitochondrial dysfunction, energy imbalance and sequential activation of autophagy and apoptosis in Sertoli cells has recently been demonstrated by our research group [44]. In line with this, in a rat model, tebuconazole induces sperm abnormalities, hormone alterations, lipid peroxidation, protein oxidation, and severe DNA degradation, leading to testis impairments [45]. Also, the ability of the endocrine disruptor vinclozolin to disrupt spermatogenic output and compromise male fertility has recently been demonstrated [46]. In this regard, the importance of investigating the effects of endocrine-disrupting mixtures in their risk assessment has recently been reported [47]. Indeed, cocktail effects and cooperative interactions of chemical mixtures are of major concern to international public and regulatory authorities.

On this basis, in the present study we have evaluated four commonly used azoles of emerging concern (PCZ, MCZ, KCZ, and FCZ; Figure 1) to assess their individual and combined potential in inducing in vitro toxicity, redox status alterations, and OS in mouse TM4 SCs. To this end, we explore the effects induced by combinations of PCZ (extensively used in agricultural and industrial applications) with MCZ, KCZ, and FCZ (commonly used individually for the treatment of surface and deep mycoses). Therefore, assessment of drug-drug interactions and flow cytometric evaluation of both cell cycle distribution and cell death were determined. Given the strong relationship between ROS and apoptosis [48], changes in intracellular ROS levels, GSH homeostasis, and the enzymatic antioxidant defence (superoxide dismutase, catalase, GSH peroxidase, GSH reductase, and GSH-S-transferase) were investigated. Finally, the eventual contribution of ROS to the azoles-induced apoptosis was examined using a well-known ROS inhibitor, N-acetylcysteine (NAC), a precursor of intracellular GSH.

## 2. Results

### 2.1. Synergistic Antiproliferative Effects of the Combined Treatment with PCZ, KCZ, and MCZ in TM4 Cells

At 48 h of incubation, the non-linear fitting of the dose-response curve of the drugs alone was used to estimate the inhibitory concentration at three different levels of efficacy: *IC*_20_, *IC*_50_, and *IC*_80_ (Table 1), which respectively identify the 20%, 50%, and 80% fractions of cells affected by the drugs (*Fa*). The half-inhibitory concentration, *IC*_50_, ranged from about 34 µM for MCZ alone, to about 73 µM for PCZ and KCZ. 

The interactions between antifungal PCZ and antimycotic drugs MCZ and KCZ were ascertained by *chequerboard assays*, where 77 combinations of drugs can be performed in a 96-well microplate; the 2D-surface graphs obtained were represented in Figure 2A,B for PCZ/KCZ and PCZ/MCZ combinations. The Interaction Index (*II*) was then calculated as described in the Materials and Methods paragraph, allowing us to summarise and quantify the interactions of PCZ/MCZ and PCZ/KCZ in terms of synergism, antagonism, or simply additivity to the fraction of cells affected (*Fa*) by the combined treatments. Out of 77 combinations, 29 were found to be synergistic for PCZ/MCZ, in addition to 30 for the PCZ/KCZ combinations (*II* values lower than 1) (Figure 2C,D). 

To further characterise the effects of the antifungal and the antimycotic compounds on cell viability and functionality, the combinations with a *II* < 1 and an *Fa* value around 0.2 were chosen. This condition is satisfied at concentrations around the *IC*_20_ values estimated for each compound alone. 

### 2.2. PCZ/KCZ Combination Enhances the G0/G1 Phase Arrest in TM4 Cells

In order to more deeply investigate the azoles-induced antiproliferative effect, analyses of the cell cycle phases distribution were performed using a flow cytometer. Azoles differentially impact cell cycle distribution in a time-dependent fashion. Specifically, at 24 h treatments, the PCZ/KCZ combination significantly increased the percentage of cells in the G0/G1 phase (63 ± 4.2) when compared to the vCTR (40 ± 2.8); this effect was accompanied by significant decreases in the S phase (PCZ/KCZ: 29 ± 2.1; vCTR: 50 ± 4.7). No remarkable differences were found for other treatments. In addition, at 48 h of incubation, the induction of G0/G1 phase arrest increases to become significant for all azoles treatments (alone and mixtures; Figure 3), with a more marked effect still detectable in the PCZ/KCZ mixture (PCZ/KCZ: 70 ± 2.6; vCTR: 46 ± 1.5). 

### 2.3. PCZ Enhances KCZ- and MCZ-Induced Apoptosis in TM4 Cells

In order to inquire if apoptosis was a mechanism of azoles-induced cell death, analyses performed by flow cytometry using annexin V/PI staining revealed differential levels of apoptosis in both combined and single compound treatments. While a level of apoptosis-induction occurred in KCZ and MCZ single treatments (48 h; Figure 4A,B), no significant differences in samples treated with PCZ alone were found. Instead, a synergistic effect, already proposed by the computational methods, found acceptance in the more marked apoptosis-induction for both the mixtures (Figure 4A,B). As a further confirmation, we measured the levels of 89-kDa Poly-(ADP-ribose) polymerase 1 fragment (C-PARP1). In the context of apoptosis, PARP1 is cleaved by caspase 3, resulting in an 89-kDa C-PARP1 protein. In agreement with previous results, C-PARP1 protein levels were significantly increased in KCZ, MCZ, and their combinations with PCZ-treated cells (Figure 4D).

### 2.4. Early Increased ROS Production in Azoles-Induced Apoptosis

Recently, the ability of azoles to induce OS in different cellular models was reported. Thus, to investigate whether azoles induced ROS accumulation, SCs were exposed to PCZ, KCZ, MCZ, and their combinations for 30 min and 3 h, and then analysed for DCFH-DA fluorescence. As shown in Figure 5B, after 3 h of incubation, azoles elicited pro-oxidant properties observed as a significant increase in ROS production (PCZ = ~1.7-fold increase; KCZ = ~1.7-fold increase; MCZ = ~1.4-fold increase; PCZ-KCZ = ~2.0-fold increase; PCZ-MCZ = ~1.6-fold increase). This effect was already detectable following 30 min of incubation, although to an overall lesser extent (Figure 5A). 

Considering the strong relationship between ROS and cell death, the contribution of ROS to azoles-induced apoptosis was also investigated. Co-incubation for 48 h with 1 mM antioxidant N-acetylcysteine (NAC) significantly reduced ROS generation (data not shown) along with KCZ- and PCZ/KCZ-induced apoptosis and necrosis (Figure 4C), suggesting a ROS-dependent mechanism of cell death.

### 2.5. Mitochondrial Membrane Potential (ΔΨ_m_) Loss in Azoles-Treated Cells

In order to evaluate if the azoles-induced ROS was associated with changes in ΔΨ_m_, the mitochondrial activity was assessed by staining with JC-1 dye. Specifically, we determined the percentage of cells with ΔΨ_m_^high^ and the mean intensity of red-orange fluorescence (MFI) emitted by each cellular sample. As shown in Figure 6A,C, azole derivatives, alone or in combination, could induce different levels of depolarisation of the inner mitochondrial membrane in SCs. In particular, at 3 h of incubation, the percentage of cells with ΔΨ_m_^high^ significantly decreased in all azoles-treated groups with respect to vCTR. However, in PCZ, KCZ, and in mixtures, this effect was also associated with increases in MFI (Figure 6B,C). 

### 2.6. Azoles Exposure Drastically Reduces SOD Activity Levels and Down-Regulates the Intracellular GSH Pool

In order to further investigate the azoles-induced alterations in cellular redox homeostasis, assays for measuring the enzymatic and non-enzymatic antioxidant activities were performed. In general, as shown in Figure 7, all treatment conditions overall affected the activities of the antioxidant enzymatic pattern in SCs. In particular, at 3 h of incubation, a strong reduction of the total SOD specific activity in all azoles-exposed samples was observed compared to control groups. Nevertheless, CAT-specific activity showed higher values with respect to the untreated counterpart, although only PCZ/KCZ resulted significantly. Generally, a positive and significant azole-induced effect in the GST activity was also found with the only exception being the PCZ-alone treated sample that was substantially reduced compared with controls. By contrast, no significant effects in GR and GPx specific activities were detected. Overall, the marked changes in SOD-specific activity indicate a reduced efficacy of the enzymatic antioxidant system. In this regard, the decrease in the SOD/(GPx + CAT) ratio (vCTR = 0.46 ± 0.067; PCZ = 0.0972 ± 0.019; KCZ = 0.178 ± 0.038; MCZ = 0.142 ± 0.015; PCZ/KCZ = 0.112 ± 0.010; PCZ/MCZ = 0.174 ± 0.036) also confirmed a decline in O_2_^−^ scavenging ability. Consistent with the increased production in ROS, azoles induced significant alterations in the GSH system in the form of significant intracellular GSH depletion in azoles-treated cells found (Figure 8). Therefore, a negative shift in both the cellular GSH/GSSG (Figure 8C) redox balance and GR/(GPx + GST) ratio, although not significant, was observed (Figure 7), also suggesting the presence of an OS condition as well as GSH recycling deficiency. However, the treatment with PCZ alone deserves a separate mention, since better efficiency in maintaining GSH homeostasis than other treatment conditions was observed. In this context, only a slight decrease in the GSH content was associated with a positive and significant shift in the GR/(GPx + GST) ratio.

## 3. Discussion

SCs are the target of numerous toxicants and are consequently a widely used cellular model for toxicity investigations in the male reproductive system [49]. They play a critical role in orchestrating the process of testis development and spermatogenesis. Thus, toxicant-mediated impairment of SCs might lead to male subfertility/infertility [50,51]. An emerging mechanism contributing to male reproductive disorders by environmental contaminants is related to ROS accumulation and redox imbalance in the testis [13,52]. 

In the present study, we demonstrate that antifungal compounds KCZ and MCZ, alone or in synergistic combination with PCZ, induce a marked degree of toxicity in mouse TM4 SCs. In this context, cytotoxicity is associated with inhibition of cell proliferation, G0/G1 cell cycle arrest, mitochondrial dysfunction, OS, and apoptosis. By itself, PCZ has cytostatic and pro-oxidant properties, but it is not able to induce cell death. In comparison, FCZ, either individually or in combination, shows no cytotoxic effects up to 320 µM. Our findings also indicate that apoptosis is correlated with increased ROS generation and an imbalance of enzymatic antioxidant activities. In this sense, inhibition experiments confirm a direct involvement of ROS-dependent mechanisms in mediating KCZ- and PCZ/KCZ-induced apoptosis. 

Our results are in line with those previously reported [52]. In particular, OS induced by many endocrine disruptors, including bisphenol A (BPA), is linked to male infertility [18]. Phthalates can also induce increased OS, acting directly on SOD, CAT, and GPx, which modulate ROS generation [53,54]. In turn, these effects can result in lipid peroxidation, destroying germs and SCs [55]. The xenoestrogen 4-nonylphenol (NP) has also been found to induce apoptosis in testicular SCs [56,57,58]. In particular, exposure of SCs to NP induces decreased cell viability, G2/M arrest, ΔΨ_m_ loss, and increased ROS production causing apoptosis and necrosis [58]. Also, human SCs exposed to monobutyl-phthalate (MBP) exhibit OS and apoptosis [59]. Very recently, it has been shown that exposure of TM4 cells to polycyclic aromatic hydrocarbons (PAHs), including benzo(a)pyrene (BaP), pyrene (Py), fluoranthene (Fl), and phenanthrene (Phe) elicited BTB disruption and apoptosis, involving OS and mitochondrial dysfunction [60].

In an attempt to transpose in vitro investigations with in vivo results, it becomes interesting to juxtapose the azole concentrations investigated in our work with those detected in human fluids. Thus, KCZ and MCZ *IC*_20_ (KCZ, 18.3 µM; MCZ, 8.6 µM) lie well within the physiological range. These two compounds allow the treatment of systemic and local infections, and specifically, KCZ is used for suppressing cortisol production in the medical management of Cushing’s disease. In this latter case, dosages of up to 1200 mg daily might be applied [61]. Although it has been demonstrated that there is a little systemic absorption of topical KCZ [62] after treatment with 400 mg/day therapy, the plasma concentration of KCZ in human subjects is 24.89 µM [63]. Concerning MCZ, the pharmacokinetic parameters were determined in human plasma after intravenous infusion at a dose of 522 mg, revealing C_max_ values ranging from 4.8 to 21.5 µM (15 min after the infusion) [64]. On the contrary, no literature data reports the presence of PCZ in human fluids. However, moderate bioaccumulation (PCZ residues of ≥ 0.01 mg/kg in edible tissues) in cattle has been reported [65].

Early studies relegate mitochondrial dysfunction to play a critical role in KCZ-induced cytotoxicity [66,67]. More recently, other authors have shown that mitochondrial toxicity of KCZ is associated with superoxide anion production and the expression of SOD2 mRNA in HepG2 cells [39]. In agreement with these findings, KCZ, MCZ, and PCZ can affect ΔΨ_m_ in SCs, with mixtures showing the highest potential compared to single compounds. The mechanism underlying the azole-induced decrease in mitochondrial activity may involve changes in mitochondrial membrane fluidity. In this regard, it has been hypothesised that, as a result of their lipophilic nature, azoles may accumulate in the mitochondrial membrane and interfere with energy-producing reactions [44]. Thus, according to their high log K_ow_ (log octanol-water partition coefficient) KCZ (log K_ow_ = 4.35) and MCZ (log K_ow_ = 6.35) have a strong potential for membrane permeation and bioaccumulation [19]. On the other hand, mitochondria are not targeted by FCZ, whose hydrophilic nature is associated with a very low partition coefficient value (log K_ow_ = 0.5). Alternatively, depolarisation of the mitochondrial membrane may be associated with changes in cholesterol content, given the ability of azoles to inhibit mammalian CYP51 and thus interfere with lipid metabolism.

As a matter of fact, alterations in mitochondrial cholesterol levels are associated with mitochondrial dysfunction and reduced antioxidant levels [68]. Furthermore, azoles have been listed in ToxCast^TM^ libraries as chemicals affecting de novo cholesterol synthesis, leading to neurodevelopmental toxicity [69]. ΔΨ_m_ is a key indicator of mitochondrial energy states, and is fundamental for maintaining cellular viability and redox homeostasis [70]. A drop in ΔΨ_m_ can lead to ROS accumulation and mitochondrial dysfunction, a possible prerequisite for inducing apoptotic or necrotic cell death. In line with these findings, loss of ΔΨ_m_ induced by azoles in SCs is associated with increasing ROS generation and apoptosis induction. The behaviour of combinations also shows major activity in this case, while PCZ alone appears to have a weak effect. The finding reinforces the relationship between ROS and apoptosis; the antioxidant NAC protects against KCZ- and PCZ/KCZ-induced apoptosis, as demonstrated by cellular growth and viability increases. Notably, loss in ΔΨ_m_ also occurs with concomitant hyperpolarization (high MFI values) in undamaged mitochondria, suggesting the chance of enhanced mitochondrial fission, as we previously described [44].

Our study also reveals the antioxidant enzymatic system as a potential target of azoles, as an important effect of deregulation has been observed. Therefore, in addition to mitochondrial ROS overproduction, azoles-induced OS may result from a reduced clearance of free radicals by the antioxidant defence system. SCs are equipped with efficient antioxidant defence and redox systems against OS. The major enzymatic components include SOD, CAT, GPx, GR, and GST. SOD is responsible for the dismutation of the toxic superoxide anion (O^2−^) to hydrogen peroxide, which either CAT or GPx subsequently removes. Thus, the presence of this enzymatic triad ensures an effective scavenging action against ROS. Our data indicate that azoles-triggered ROS induction occurs in parallel with a marked and significant decrease in SOD-specific activity, as well as an increase in CAT-specific activity, although not significant (except for PCZ/KCZ). These changes cause a reduction in the SOD/(GPx + CAT) ratio, thus indicating a marked lowering in O^2−^ scavenging ability. If not quickly removed and/or neutralised, the enhanced and prolonged generation of O^2−^ might lead to OS-promoting conditions and genotoxic effects. In fact, regardless of its low chemical reactivity, O^2−^ is able to cause significant cell injury. In particular, superoxide has been involved in generating mitochondrial/nuclear DNA damage by promoting hydroxyl radical formation and/or accelerating iron release from storage proteins or [4Fe-4S] clusters [71,72]. Furthermore, in a scenario characterised by reduced SOD activity, the increased levels in CAT activity may reflect H_2_O_2_-dependent cytotoxicity originating from CYPs-mediated catalytic uncoupling [73]. Therefore, this condition may occur to a greater extent in the PCZ/KCZ paradigm.

The GSH system is the main regulator of redox homeostasis, and represents the most important cellular antioxidant against oxidative damage. A decrease in GSH content has long been demonstrated to be a preliminary episode in the OS-induced apoptosis process [74]. In particular, in numerous cell models it has been reported that GSH depletion results in a drop of ΔΨ_m_, inactivity of the complex of the respiratory chain, an increase in ROS accumulation, and liberation of apoptogenic factors [75]. In agreement with these findings, under our experimental conditions the presence of azoles (except for PCZ) induced a remarkable reduction of intracellular GSH levels, indicating a modification of the cellular redox status. Also, a negative shift in the GSH/GSSG ratio, albeit not significant, was recorded. Changes in the redox balance of GSH depend on GSH export, GSH/GSSG recycling, as well as GSH synthesis. Furthermore, GSH is also required as a co-substrate of antioxidant enzymes. In this respect, azoles treatment induces positive and significant effects in the specific activity of GST, without modifying GR activity. Thus, GSH depletion may partly be attributed to the azoles-induced changes in GSH recycling efficiency, since the GR/(GPx + GST) ratio is slightly decreased. In the PCZ scenario, differently from single treatments with KCZ and MCZ, the lower production of ROS combined with improved efficiency in maintaining GSH redox homeostasis may explain the absence of apoptosis induction. 

These findings may be important, considering the crucial and functional role played by SCs in supplying spermatogenic cells with GSH by direct interaction [76]. GSH is the storage form of cysteine (Cys), and γ-glutamyl transpeptidase (GGT) in the germ cell membrane metabolises GSH to individual amino acids. The released Cys is then used by the spermatogenic cells for protamine biosynthesis. On the other hand, data on GGT-deficient mice showed reduced seminal vesicle and testis sizes associated with severe oligozoospermic and infertility conditions [77]. Accordingly, the GSH supplementation from SCs is essential as a defence against ROS and as a source of cysteine, useful for spermatogenesis. In this sense, intracellular GSH may be considered a biomarker of SC function (as Cys donor). 

In conclusion, our results have revealed the mitochondrial network and antioxidant enzymatic and non-enzymatic systems as potential primary targets of azoles-induced toxicity in TM4 cells. Although our in vitro findings need to be replicated by in vivo models, given the background described above, azoles may have a central role in the negative impact of OS on male fertility, as well as on the regulation and cross-talk between SCs and sperm cells [13]. In this regard, to elucidate the role of ROS, the potential involvement of p38/MAPK signalling in mediating azole-induced apoptosis should be investigated. Therefore, further studies are required to assess the potential risk of the pervasive exposure on humans of these multiple chemical insults. 

## 4. Materials and Methods

### 4.1. Cell Culture and Treatments

The TM4 mouse Sertoli cells (ATCC^®^ CRL1715™) were obtained from the American Type Culture Collection (Sigma-Aldrich, St. Louis, MO, USA), and were routinely cultured at standard conditions (37 °C in a 5% CO_2_ humidified atmosphere), seeded at a density of 1 × 10^4^ cells/cm^2^, maintained in DMEM (Dulbecco’s modified Eagle medium)/ Ham’s F-12 50/50 Mix with 15 mM HEPES (Corning Life Sciences, Manassas, VA, USA); they were supplemented with 2 mM glutamine, 2.5% heat-inactivated fetal bovine serum (FBS) and 5% heat-inactivated horse serum (HS) (EuroClone, Pero, MI, Italy), penicillin 100 IU/mL, and streptomycin 100 µg/mL (Corning Inc., Somerville, MA, USA), until they reached a confluence close to 80%. Cells were maintained in standard conditions for 24 h before treatments. For experiments with azoles, cells were then exposed to PCZ, KCZ, MCZ, FCZ (≥98% purity, from European Pharmacopoeia, Council of Europe, Strasbourg, France), and mixtures at the final concentrations, and for the incubation times indicated below. All azoles were dissolved in DMSO. In an effort to avoid a possible experimental artifact with vehicles, experiments were performed including control groups in which the effect of the drug vehicle alone was tested (vehicle control, vCTR). In these experimental conditions, we excluded any negative effects of DMSO that is biocompatible with no biological effect per se. For experiments with N-acetylcysteine (NAC), 24 h after seeding cells were pre-treated for 1 h with the ROS inhibitor (1 mM) and then maintained in co-treatment with azoles for 48 h.

### 4.2. Cell Growth and Viability

TM4 cells were seeded at a density of 1 × 10^4^/cm^2^, and cultures incubated at standard conditions in the presence or the absence of azoles to reach final concentrations ranging from 0.078 µM up to 320 µM in culture medium. Cell growth and viability were assessed at 24 and 48 h by Trypan-blue exclusion assay and MTT (3-(4,5-dimethylthiazol-2-yl)-2,5-diphenyl-2H-tetrazolium-bromide) colorimetric method (Sigma-Aldrich, St. Louis, MO, USA), quantifying the metabolic efficiency of living cells. The MTT was used to seed cells on a 96-well plate in order to evaluate the effects on cell viability of the azoles alone and in combination, along with the interaction degrees by analysing the surface response based on the Bliss independence (BI) and Chou and Talalay models, using the *chequerboard assay* as previously described [44]; briefly, cells were treated for 24 and 48 h with increasing concentrations in azoles, ranging from 0.078 µM (for PCZ) and from 1.25 µM (for FCZ, KCZ and MCZ) to 80 µM for KCZ and MCZ, and up to 320 µM for FCZ, both alone and in binary combinations. The 2D-surface graphs obtained were represented in Figure 2A,B, for PCZ/KCZ and PCZ/MCZ combinations. 

### 4.3. Calculation of the Interaction Index

The Interaction Index (*II*) was calculated, as previously described [44], by modifying the Chou and Talalay equation [78]. Specifically, Chou and Talalay’s median-effect method estimates the *IC*_50_ and the sigmodicity coefficient *m* values using linear regression of the linearised dose-response curve. Greco et al. [79] suggested that the *IC*_50_ and *m* values can be more comfortably obtained using non-linear regression of the Hill equation.

For a two-drug assay, *II* is the summation of the ratios of the doses (*D_n_*) of the drugs in combination, and the concentrations of the two drugs alone (*ID_x,n_*), to obtain a fraction *x* of cells affected by drug combinations (*Fa*). *ID_x,n_* is estimated by the following equation:(1)IDx,n=IC50(Fa1−Fa)1mn

*II* is calculated and plotted against the *Fa* value for each drug combination by generating the *Fa*-*II* plot. *II* values < 1 suggest synergism between drugs, *II* values > 1, antagonism, while for *II* equal 1, additivity is called.

Results from the dose-response curve and drug-drug interaction assays were analysed by the software OriginPro ver. 8.5 (OriginLab Corporation, Northampton, MA, USA).

### 4.4. Flow Cytometry Analyses of Cell Cycle

Control (vCTR) and treated cells were collected and subsequently washed twice with ice-cold PBS, then fixed by using a cooled 70% ethanol solution (Sigma-Aldrich, Saint Louis, MO, USA) in PBS, with gentle mixing at 4 °C for 30 min. Fixed cells (10^6^ cells/mL) were washed twice with ice-cold PBS, and resuspended with a solution containing 50 µg/mL PI, 0.1% Nonidet-P40 and RNase A (6 µg/10^6^ cells) (all reagents acquired from Sigma-Aldrich, Saint Louis, MO, USA) for 30 min in the dark at 4 °C. Data from 10,000 events per sample were collected and analysed using an FACS Calibur instrument (Becton Dickinson (BD) Instruments Inc., San José, CA, USA ) equipped with cell cycle analysis software (Modfit LT for Mac V3.0), in order to calculate the percentages of cells in the G1, G2/M, and S phases.

### 4.5. Detection of Intracellular Reactive Oxygen Species (ROS)

The intracellular ROS production was detected by using 2′,7′-dichlorofluorescein diacetate (DCFH-DA) (Molecular Probes, Eugene, OR, USA), as previously reported [80]. In brief, after 30 min and 3 h of treatments, samples were incubated with 1 µM DCFH-DA at 37 °C for 30 min. Subsequently, cells were collected and twice washed in ice-cold PBS; samples were then analysed using flow cytometry to detect the presence of intracellular ROS at the wavelengths of 502 and 524 nm, for the excitation and the emission, respectively; the fluorescence intensity was detected with a Perkin Elmer LS-50B spectrofluorometer. Cells treated with 500 µM tert-butyl hydroperoxide (t-BHP) for 1.5 h were used as a positive control for ROS generation. 

### 4.6. Enzymatic Activity Assays

The antioxidant enzymatic activities were performed spectrophotometrically, as previously described [81]. Briefly, after 3 h of incubation with azoles, cells were washed in ice-cold PBS, resuspended in buffer containing 50 mM Tris–HCl (pH 7.4), 1% (*v/v*) Triton X-100, 1 mM EDTA, and then disrupted by three freeze-thawing cycles. After centrifugation (17,000 g, 10 min, 4 °C), the supernatants were collected for protein quantification and subsequently analysed.

The glutathione reductase (GR) activity was evaluated spectrophotometrically by measuring NADPH oxidation (340 nm at 25 °C). The reaction mix contained 50 mM potassium phosphate buffer (pH 7.0), 1 mM GSSG, 0.2 mM NADPH, 1 mM EDTA, and the adequate protein amount.

Total glutathione peroxidase (GPx) activity (se-dependent and se-independent) of each cell extract was assayed spectrophotometrically by following the NADPH oxidation (340 nm). The assay mixture contained 50 mM potassium phosphate buffer (pH 7.0), 1 mM GSH, 0.01 U/mL GSH reductase, 1 mM EDTA, 0.2 mM NADPH, 70 µM t-BHP, and the adequate amount of the protein extract. The glutathione transferase (GST) activity was evaluated by determining the GSH conjugation rate to 1-chloro-2,4- dinitrobenzene at 340 nm. The reaction mixture was prepared with 0.1 M potassium phosphate buffer (pH 6.5), 2 mM GSH, 1 mM 1-chloro-2,4-dinitrobenzene, and a determined amount of the supernatants. 

The catalase (CAT) activity was evaluated, by monitoring at 240 nm its 10-millimolar decomposition. One unit was defined as 1 µmol of H_2_O_2_ reduced/min at 25 °C.

The superoxide dismutase (SOD) activity was determined using a colourimetric activity assay (ThermoFisher Scientific—Life Technologies Corp, Carlsband, CA, USA). The assay was conducted according to manufacturer’s instructions; briefly, it measures all types of SOD activity (including Cu/Zn, Mn, and FeSOD). Samples were resuspended in specific coloured diluent and added to a 96-well microplate. After the addition of the substrate, samples were subsequently incubated with xanthine oxidase reagent (room temperature for 20 min). In the presence of oxygen, the xanthine oxidase produces superoxide, which converts a colourless substrate (in the detection reagent) into a yellow-coloured product, revealed at 450 nm length. Increasing levels in SOD correspond to a decrease in superoxide concentration and a less yellow product. SOD-specific activity was expressed as U/mg protein, while the specific activity of all the other enzymes analysed was expressed as nmol/min/mg protein.

### 4.7. Detection of GSH and GSSG Intracellular Content

GSH and GSSG intracellular contents were detected by using a colourimetric assay (ThermoFisher Scientific—Life Technologies Corp, Carlsband, CA, USA). The analyses were performed according to the manufacturer’s instructions after treatments with azoles at 3 h of incubation; in brief, cell lysates were prepared in ice-cold 5% 5-sulfo-salicylic acid dehydrate (Sigma-Aldrich, St. Louis, MO, USA), and subsequently resuspended in the supplied diluent before adding them into a 96-well microplate. After promptly adding substrate, the reaction with the free thiol group of GSH initiates, producing a highly coloured product. The relative absorbance was detected at 405 nm. Assays were performed at least in triplicate for the GSH and the GSSG content detection. To determine oxidised glutathione content, 2-vinylpyridine (2-VP) (Sigma-Aldrich, St. Louis, MO, USA) was added to block free GSH or other thiols contained in each sample; the specific concentration of the GSSG was determined by performing a 2-VP-treated samples standard curve.

### 4.8. Assessment of ΔΨ_m_ by Flow Cytometry

Variations of the ΔΨ_m_ in samples exposed to azoles for 3 h were assessed using the lipophilic cation dye JC-1, as previously reported [44,82]. Cells (1 × 10^6^ per sample) were stained for 30 min at 37 °C in a humidified atmosphere, with 3 µM JC-1 (Molecular Probes, Eugene, OR, USA) and collected; after washing in PBS, cells were analysed using flow cytometry. To assure a positive control for the abatement of the ΔΨ_m_, the cells were cultured in the presence of 25 mM rotenone (Sigma-Aldrich, St. Louis, MO, USA) for 1 h at standard conditions, and then incubated with the potentiometric dye as described above. Experiments were performed in triplicate and analysed by an FACS Calibur instrument (BD Instruments Inc., San José, CA, USA). The fluorescent signals from JC-1 monomers or aggregates were detected through the FL-1 (525 ± 5 nm) and FL-2 bandpass filters (575 ± 5 nm), respectively; the forward and side scatter channels gated a minimum of 1 × 10^4^ cells on the major population of normal-sized cells. ΔΨ_m_ data assessment was performed by the Cell Quest software (BD Instruments Inc., San José, CA, USA). 

### 4.9. Annexin V-FITC and Propidium Iodide Assay

Treated and control cells (vCTR) (1 × 10^6^) were collected; after washing in ice-cold PBS, samples were resuspended in 1 mL of binding buffer (10 mM HEPES, 140 mM NaCl, and 2.5 mM CaCl2, at pH 7.4) containing 10 µg/mL Annexin V-FITC and 1 µg/mL PI. After staining (1 h incubation at room temperature), cells were then washed and subsequently analysed using an FACS Calibur instrument (BD Instruments Inc., San José, CA, USA). Cells treated with 25 µM Etoposide (Eto) for 24 h were used as a positive control for apoptosis induction. Data from 10,000 events per sample were collected and analysed using an FACS Calibur instrument (BD Instruments Inc., San José, CA, USA ) equipped with cell cycle analysis software (Modfit LT for Mac V3.0) in order to calculate the percentages of apoptotic and necrotic cells.

### 4.10. Western Blot Analysis

Total proteins were extracted from treated and control cells (vCTR) after 48 h of incubation with azoles using a lysis buffer containing 10 mM HEPES at pH 7.2, 142 mM KCl, 1 mM EDTA, 5 mM MgCl_2_, 1 mM EDTA, 1 mM PMSF, and a cocktail of protease inhibitor (Sigma-Aldrich, St. Louis, MO, USA). The protein extracts were run on a 10% SDS-PAGE and transferred onto PVDF filters and maintained at room temperature (RT) for 2 h in 5% non-fat dry milk diluted in TBST (containing 0.1% Tween-20). After washing, membranes were incubated O/N at 4 °C with the primary antibody anti-PARP1 (Cell Signaling, Danvers, MA, USA) diluted to 1:1000 in 1% non-fat dry milk in TBST. After washing, PVDF filters were incubated for 1 h at RT with the anti-rabbit (1:5000) HRP-conjugated secondary antibody (Immunological Sciences, by Società Italiana Chimici, Rome, Italy). ECL West Pico Plus chemiluminescent substrate was used to detect signals using a ChemiDoc XRSplus imaging system (Bio-Rad Laboratories, Hercules, CA, USA). The optical densities of blot bands were quantified using ImageJ software (US National Institutes of Health, Bethesda, MD, USA); the graph values represent the ratios of cleaved PARP1 to total PARP1. ß-actin was used as loading control.

### 4.11. Statistical Analyses

All analyses were performed at least in three independent experiments. Values reported in this study are expressed as the mean ± standard error (SE), unless otherwise indicated. The Sigma Stat 2.03 (SPSS, Chicago, IL, USA) was utilized to evaluate the statistical significance of differences between group means. The comparisons between multiple groups were performed using an ANOVA test, followed by Dunnett’s or Holm–Sidak methods. ANOVA on ranks (Kruskal–Wallis test) was used to analyse NAC effects on apoptosis reduction. A value of *p* < 0.05 versus vCTR was considered statistically significant.

## Figures and Tables

**Figure 1 ijms-23-05429-f001:**
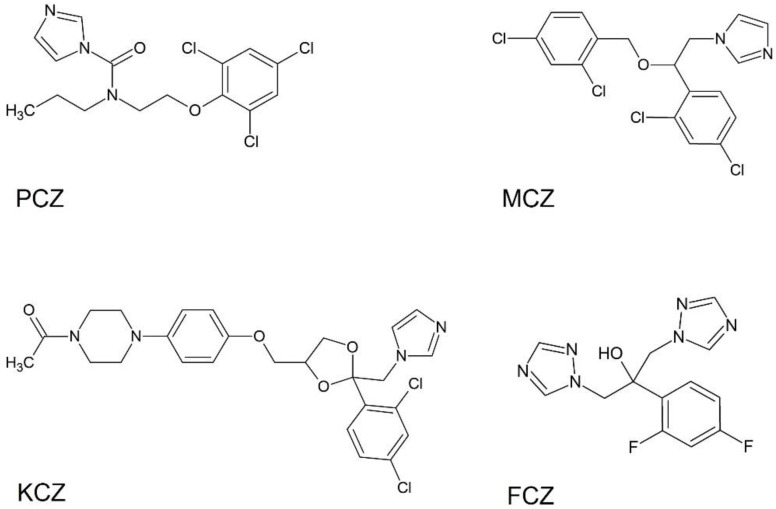
Chemical structures of the investigated antifungal azoles: prochloraz (PCZ), miconazole (MCZ), ketoconazole (KCZ), and fluconazole (FCZ).

**Figure 2 ijms-23-05429-f002:**
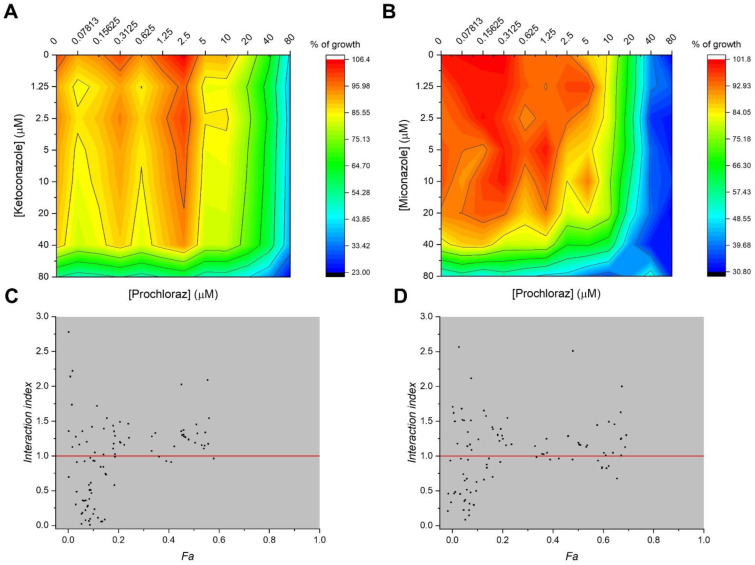
Synergistic interactions of PCZ with KCZ and MCZ in reducing cell viability and proliferation in TM4 cells. (**A**,**B**) Graphs representing the 2D-surface response ∆E model (ΔE = E_predicted_–E_measured_) and the experimental cell growth and viability obtained from the two-drug *chequerboard assay* using the MTT method in 96-well plates, for the binary combinations of PCZ/KCZ (**A**) and PCZ/MCZ (**B**). (**C**,**D**) Analysis of the combination index (CI) for PCZ/KCZ (**C**) and PCZ/MCZ (**D**); CI < 0 indicates a synergistic effect. Data represent the means of three independent experiments ±SE.

**Figure 3 ijms-23-05429-f003:**
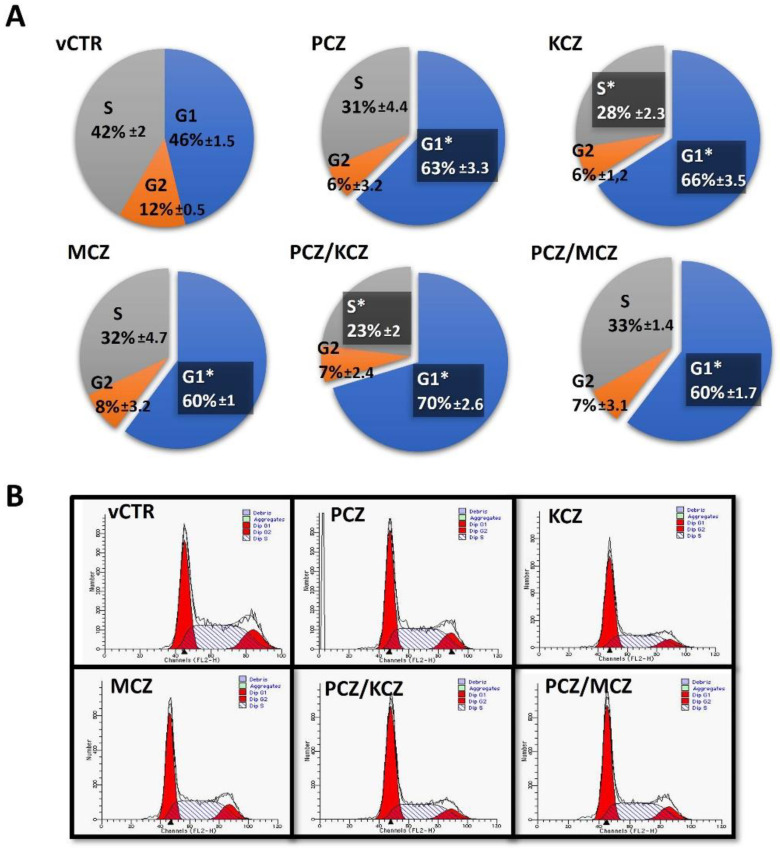
Effects of azole exposure on the distribution of cell cycle phases in TM4 cells. Cells were treated with PCZ (18.3 µM), KCZ (18.6 µM), MCZ (8.6 µM), and their mixtures for 48 h, and compared to the vCTR. The distribution of cell cycle phases was assayed using PI staining and flow cytometry analyses. The percentage of cells in each phase is presented at 48 h of incubation (**A**), and represents the mean ± SE of three independent experiments; one-way ANOVA followed by Dunnett’s test; * *p* < 0.05 compared to the vCTR. (**B**) Representative images of flow cytometric profiles after 48 h of incubation in the absence (vCTR) or in the presence of azoles.

**Figure 4 ijms-23-05429-f004:**
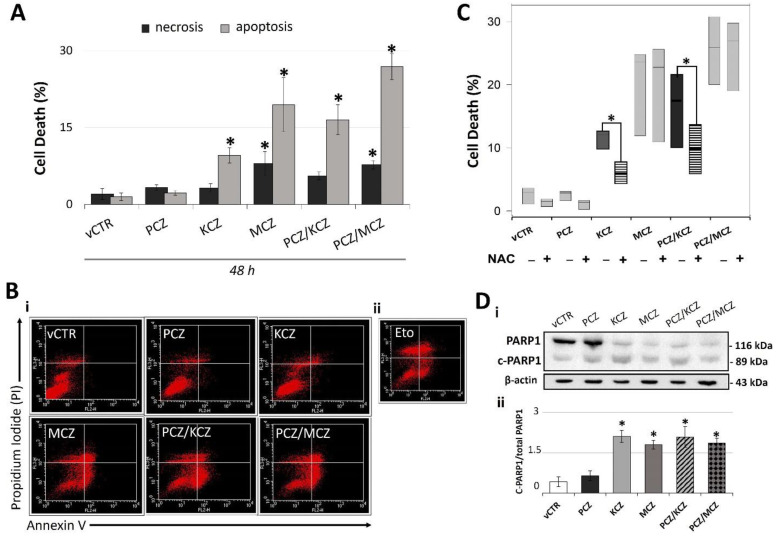
Possible ROS-dependent apoptotic activity of azole drugs in TM4 cells. (**A**) Graph representing the percentage of apoptotic and necrotic cells in experimental groups exposed for 48 h to PCZ, KCZ, MCZ, and mixtures. Data from three independent experiments are expressed as mean ± SE; one-way ANOVA followed by Dunnett’s test; * *p* < 0.05 vs. vCTR. (**B**,**i**) Representative images of cytofluorometric profiles of cells stained with annexin V and PI after 48 h of incubation in the absence (vCTR) or in the presence of azoles. (**B**,**ii**) Cells treated with 25 µM etoposide (Eto) for 24 h were used as a positive control for apoptosis induction. Upper left quadrant = necrosis; upper right quadrant = late apoptosis; lower right quadrant = early apoptosis. (**C**) Analysis of cell death (%) in samples treated or not treated (vCTR) with azoles at 48 h of incubation, in the presence (+) or in the absence (−) of the inhibitor of ROS (1 mM NAC). NAC = N-acetyl-cysteine. Values are expressed as the mean of three independent experiments ± SE; one-way ANOVA, Dunnett’s test; * *p* < 0.05 vs. vCTR. (**D**,**i**) Representative images of C-PARP1 Western blotting and (**D**,**ii**) protein expression levels of C-PARP1 (89 kDa) at 48 h exposure to azoles. Band density was performed by using Image-J software. The bar graph represents data from three independent experiments as ratios of cleaved PARP1 to total PARP1 ± SE; ß-actin was used as loading control. one-way ANOVA, followed by Holm–Sidak method; * *p* < 0.05 vs. vCTR.

**Figure 5 ijms-23-05429-f005:**
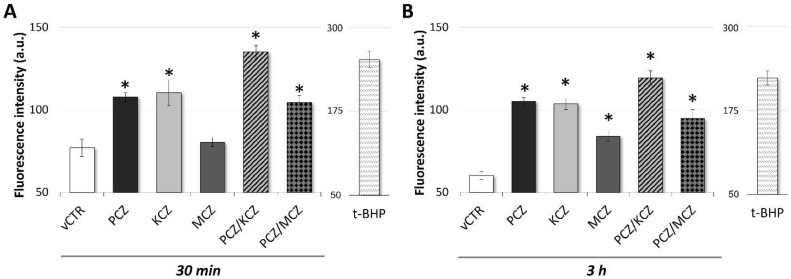
Exposure to azole drugs triggers early ROS production in TM4 cells. ROS production in TM4 cells incubated in the absence (vCTR) or in the presence of azoles fungicides, alone and in combination, for 30 min (**A**) and 3 h (**B**); intracellular ROS generation was detected by measuring DCFH-DA fluorescence with a fluorimeter, and determined in terms of fluorescence intensity (arbitrary units, a.u.); cells treated with 500 µM tert-butyl hydroperoxide (t-BHP) for 1.5 h were used as a positive control for ROS generation (A-B). Values from three independent experiments are expressed as means ± SE; one-way ANOVA followed by Dunnett’s test; * *p* < 0.05 vs. vCTR.

**Figure 6 ijms-23-05429-f006:**
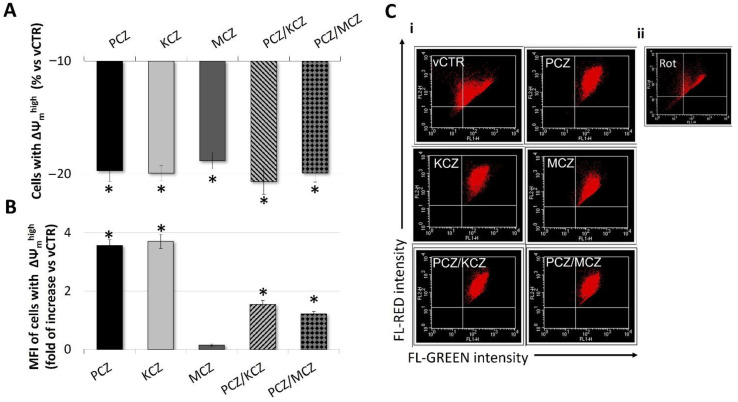
Depolarising effect of azole compounds on mitochondrial membrane potential (ΔΨ_m_) in TM4 cells. ΔΨ_m_ was measured by FACS using JC-1 (3 µM) staining. (**A**,**B**) Effect of azole fungicides on mitochondrial membrane potential in selected TM4 cells with ΔΨ_m_^high^, expressed as (**A**) the percentage of cells with ΔΨ_m_^high^ (red/orange fluorescence of JC-1 aggregates) and (**B**) MFI emitted by ΔΨ_m_^high^ populations in vCTR and azole-exposed groups (3 h). Values from three independent experiments are expressed as mean ± SE. One-way ANOVA followed by Dunnett’s test; * *p* < 0.05. (**C**,**i**) Representative images of flow cytometry profiles of JC-1-stained samples analysed at 3 h of incubation for vCTR and azole-exposed groups. (**C**,**ii**) Cells were incubated with 25 mM rotenone for 1 h as a positive control for the abolishment of ΔΨ_m_. MFI = mean red fluorescence intensity; Rot = rotenone.

**Figure 7 ijms-23-05429-f007:**
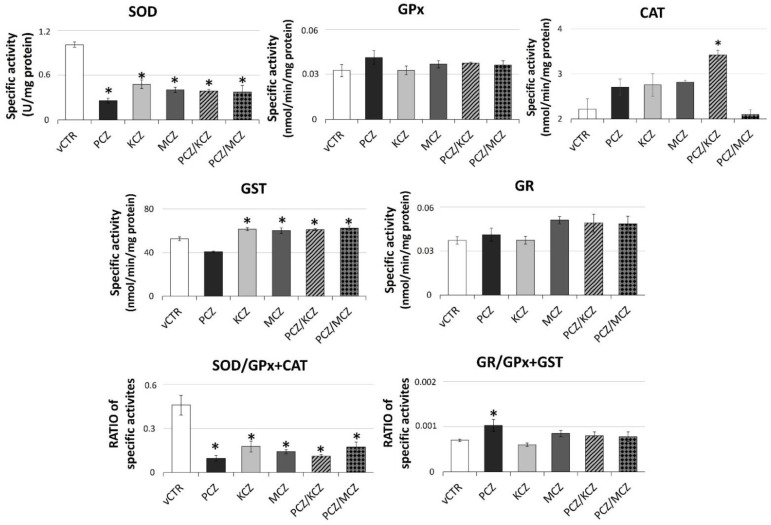
Azoles exposure significantly impairs SOD activity levels resulting in an altered SOD/GPx + CAT ratio. The specific enzymatic activities were assayed in samples incubated in the absence (vCTR) or in the presence of azoles after 3 h of exposure by spectrophotometric-based methods. Data are expressed as specific activities or ratios, as indicated, from three independent experiments and represent the mean ± SE; one-way ANOVA followed by Dunnett’s test or Holm–Sidak methods; * *p* < 0.05 vs. vCTR.

**Figure 8 ijms-23-05429-f008:**
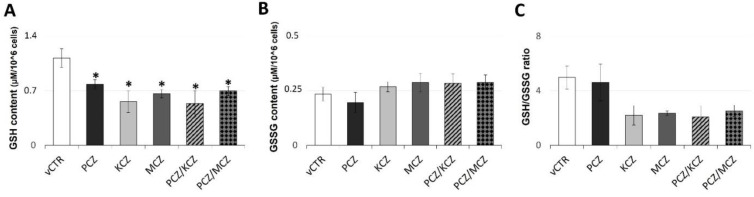
Azoles induced marked depletion of GSH reservoir with further underlying loss in GSH/GSSG ratio. Levels of (**A**) reduced (GSH), (**B**) oxidised (GSSG) glutathione, and (**C**) GSH/GSSG ratio in vCTR and azoles-treated groups after 3 h of incubation. Values from three independent experiments are expressed as mean ± SE; one-way ANOVA, followed by Holm–Sidak test; * *p* < 0.05 vs. vCTR.

**Table 1 ijms-23-05429-t001:** *IC*_20_, *IC*_50_, and *IC*_80_ of the tested azoles. The non-linear fitting of the dose-response curve of the drugs alone was used to estimate the inhibitory concentration at three different levels of efficacy: *IC*_20_, *IC*_50_, and *IC*_80_, which respectively identify the 20%, 50%, and 80% fractions of cells affected by the drugs (*Fa*) at 48 h of exposure. Values indicate the mean ± SE.

Compound	*IC*_20_ ± SE (µM)	*IC*_50_ ± SE(µM)	*IC*_80_ ± SE(µM)
PCZ	18.5 ± 3.9	73.8 ± 15.6	295.2 ± 62.3
MCZ	8.6 ± 0.6	34.4 ± 2.5	137.6 ± 9.8
KCZ	18.3 ± 2.4	73.2 ± 9.5	293.0 ± 38.1
FCZ	Not detectable		

## Data Availability

The data presented in this study are available in this paper.

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
