# Peer review of "Synergistic Activity of Ketoconazole and Miconazole with Prochloraz in Inducing Oxidative Stress, GSH Depletion, Mitochondrial Dysfunction, and Apoptosis in Mouse Sertoli TM4 Cells"

_ijms, 2022, doi:10.3390/ijms23105429_

Round 1

Reviewer 1 Report

  • The authors demonstrated the apoptotic activity of azole drugs in TM4 cells. However, only annexin V/PI staining was performed. More apoptotic assays are needed.

  • It is incorrect that the authors used CTR to represent the absence of azoles instead of vehicle control. Solvents sometimes affect many different types of cellular activity. Therefore, the authors should compare the azole-induced cellular activities with “vehicle control”.

  • The authors should explain the purpose of this study, why the authors were interested in the cytotoxic effects of different antifungal azoles combination? Are these combinations common in anti-fungal therapy?

  • In the Figure 3A, the authors showed that PCZ & KCZ alone elevates the percentage of G1 population, however, in Fig. 3B the G1 population (PCZ & KCZ) looks lower than CTR. Please clarify this issue.

  • In Figure 4B, why there is two cell population in the CTR group?

  • In the figure4C, why the authors used different statistical analysis and figure?

Author Response

We thank the reviewers for their useful comments, that strengthen our results. Please find below a point-to-point answer to their questions. We modified the text and figures to respond to all the issues and elaborated on the changes below.

All coauthors have agreed to the revisions.

REVIEWER#1

  • The authors demonstrated the apoptotic activity of azole drugs in TM4 cells. However, only annexin V/PI staining was performed. More apoptotic assays are needed.

Response: We agree with the referee's comment. We performed additional experiments to address this issue and Figure 4D, showing Cleaved-PARP1 (as important apoptotic marker) protein levels, has been added.

See revised Results, lines 178-183

  • It is incorrect that the authors used CTR to represent the absence of azoles instead of vehicle control. Solvents sometimes affect many different types of cellular activity. Therefore, the authors should compare the azole-induced cellular activities with “vehicle control”.

Response: We agree with the referee's comment. However, in preliminary experiments, we have included control groups in which the effect of the drug vehicle alone (DMSO) on TM4 cells has been tested, at concentrations up to 0.5%. Therefore, we excluded any negative effects of DMSO that results biocompatible with no biological effect per se. Notably, our IC20 working solutions do not exceed concentrations of 0.03%.

See revised MM, lines 437-441

  • The authors should explain the purpose of this study, why the authors were interested in the cytotoxic effects of different antifungal azoles combination? Are these combinations common in anti-fungal therapy?

Response: We thank the reviewer for this comment.

We modified the Introduction to respond to this issue.

  • In the Figure 3A, the authors showed that PCZ & KCZ alone elevates the percentage of G1 population, however, in Fig. 3B the G1 population (PCZ & KCZ) looks lower than CTR. Please clarify this issue.

Response: We thank the reviewer for this comment. We apologize, in the figure assembly there was a mistake. See revised Figure 3B.

  • In Figure 4B, why there is two cell population in the CTR group?

In figure 4B, the control cells are inside the low left quadrant, Ann-V negative and PI negative, being viable and healthy cells. In our experience, the TM4 cell line frequently displays a double population when untreated. However, the effects of the treatments are clearly visible, and the cell population is redistributed in the other quadrants.

  • In the figure4C, why the authors used different statistical analysis and figure?

Response: Depending on the raw data characteristics, the pre-processing statistical analyses can converge to parametric or non-parametric methods. Therefore, the Sigma plot 11.0 software analyzes raw data for Normality and Equal Variance and then it proceeds for the best analysis. In this case, the presence of NAC added a more complex elaboration level of analysis and the choice lied on non-parametric methods. Accordingly, we used the box-plot representation.

Reviewer 2 Report

The manuscript ijms-1678722 concerns the in vitro impact of ketoconazole, miconazole, and prochloraz on oxidative stress-related processes in mouse Sertoli cells. The topic is suitable to the journal. Nevertheless, several parts of the manuscript require revision in order to add clarity or to make the manuscript better focused. I have listed some concerns/suggestions below:

  1. I suggest including 1-2 sentences on the male infertility epidemiology to emphasize the purpose of the work. In my opinion, the introduction section is written quite chaotically. It would be useful to start with the importance of Sertoli cells in male fertility, the importance of oxidative status in male fertility, and finally a description of the influence of the tested substances on male fertility.
  2. In my opinion, describing the impact of these substances on fungi is unnecessary in the introduction section, authors should focus on mammals to avoid the chaos.
  3. Did the Authors perform longer exposure experiments (72, 96h)? Did they notice any change in cell viability? 
  4. Were IC values calculated after 24 or 48h of exposure?
  5. Why were these combinations selected for further analyzes? Why PCZ+MCZ or KCZ+MCZ combinations were not used in the study? I suggest to focus on studying the exposure to separate compounds and the effects of mixture of all studied compounds (and not choosing subjectively two compounds as environmental exposure does not apply to selective exposure to chosen compounds from the group).
  6. The justification of using only one cell line is needed. As the results are not validated on second cell line model, they refer only to the model used, not to Sertoli cells in general.
  7. The Authors state that the studied concentrations of azoles are similar to physiological concentrations. Could the Authors describe the human biofluid concentrations of the substances?
  8. The results section includes the interpretation that should be moved and described in the discussion section (e.g., 110-111, 154-156, 200-201).
  9. Time dependent changes for azole specific-cell cycle changes should be also included in the Figure 3 as 24 and 48h exposures were performed. Fig. 3 and the footnote do not include the direct information on the duration of the exposure.
  10. Both negative and positive control should be used (Fig. 3, 4, 5, 6 lacks positive controls to compare the effects of the azoles).  
  11. Fig. 7 – some unification of the figures should be included (the figure for catalase looks different than other figures which introduces chaos).
  12. The Authors do not show any relationship between oxidative stress and the function of the cell model used, they do not investigate the effect of azole-induced oxidative stress on Sertoli cell function. Additional studies on Sertoli cells could be performed to evaluate if oxidative stress affects the Sertoli cell function. 
  13. Discussion section lacks the discussion supported with the published literature data. The results are not comprehensively interpreted considering other compounds present in the environment, their concentrations, the importance of oxidative status, and the Sertoli cell function.
  14. Conclusions should be improved. I do not see any specific conclusions from the manuscript. Questions should be answered: Do environmental and biofluid azole concentrations affect the oxidative status in Sertoli cells? Which mechanisms are affected? How can we interpret the interactions between the azoles? Does oxidative stress affect Sertoli cell function?
  15. As a general, English editing is needed. Commas in the figures should be converted to dots.

Author Response

REVIEWER#2

The manuscript ijms-1678722 concerns the in vitro impact of ketoconazole, miconazole, and prochloraz on oxidative stress-related processes in mouse Sertoli cells. The topic is suitable to the journal. Nevertheless, several parts of the manuscript require revision in order to add clarity or to make the manuscript better focused. I have listed some concerns/suggestions below:

  • I suggest including 1-2 sentences on the male infertility epidemiology to emphasize the purpose of the work. In my opinion, the introduction section is written quite chaotically. It would be useful to start with the importance of Sertoli cells in male fertility, the importance of oxidative status in male fertility, and finally a description of the influence of the tested substances on male fertility.

Response: We thank the reviewer for this comment. We modified the Introduction to respond to this issue.

  • In my opinion, describing the impact of these substances on fungi is unnecessary in the introduction section, authors should focus on mammals to avoid the chaos.

Response: As suggested we modified the Introduction to respond to this issue.

  • Did the Authors perform longer exposure experiments (72, 96h)? Did they notice any change in cell viability?

Response: Taking into consideration the cell density parameter (1 × 104 cells/cm2), Sertoli cell reached a confluence close to 80% at 48 h of incubation. Therefore, it is not possible to perform experiments at 72 and 96 h.

  • Were IC values calculated after 24 or 48h of exposure?

Response: The IC20 values were calculated at 48 h of incubation.

See revised legend of table 1

  • Why were these combinations selected for further analyzes? Why PCZ+MCZ or KCZ+MCZ combinations were not used in the study? I suggest to focus on studying the exposure to separate compounds and the effects of mixture of all studied compounds (and not choosing subjectively two compounds as environmental exposure does not apply to selective exposure to chosen compounds from the group).

Response: As described in results section, PCZ+MCZ combination has been included in our analyses. In addition, azole mixtures have been developed combining the agro-industrial compound (PCZ) vs the pharmaceutical azoles commonly used (individually) for the treatment of surface and deep mycoses (FCZ, MCZ, and KCZ).

Please see the revised Introduction, lines 98-100

  • The justification of using only one cell line is needed. As the results are not validated on second cell line model, they refer only to the model used, not to Sertoli cells in general.

Response: We thank the reviewer for this comment.

However, we would like to remark that title and the major conclusions are referred to TM4 cellular model. In addition, our in vitro model displayed most of the expected biochemical characteristics of Sertoli cells and is an established model for investigating the division of Sertoli cells (SC) (Wang et al., 2016). Also, TM4 cell line is undoubtedly the most extensively studied Sertoli cell line which value has been proven in several studies regarding the effect of toxicants on male fertility (Ji et al., 2017 – toxicant Benzo[a]pyrene;  Liu et al., 2016 – toxicant: zinc oxide nanoparticles; Liu et al., 2014 – toxicant: nonylphenol; Ge et al., 2014a,b – toxicant: Bisphenol A; Lee et al., 2009 – toxicant: 1,3-Dinitrobenzene; Choi et la., 2012 – toxicant: nonylphenol and other). This cell line has also recently been utilized to elucidate the pharmacological roles of diosgenin as a novel therapeutic methods to treat male infertility (Wu et al., 2015).

  • The Authors state that the studied concentrations of azoles are similar to physiological concentrations. Could the Authors describe the human biofluid concentrations of the substances?

Response: Please see Discussion, lines 327-333.

  • The results section includes the interpretation that should be moved and described in the discussion section (e.g., 110-111, 154-156, 200-201).

Response: We improved the text to address this issue, sentences have been moved in discussion section.

  • Time dependent changes for azole specific-cell cycle changes should be also included in the Figure 3 as 24 and 48h exposures were performed. Fig. 3 and the footnote do not include the direct information on the duration of the exposure.

Response: Please see the revised footnote in Figure 3. Data at 24 h is significant only for PCZ/KCZ, therefore this results has been reported only in the text.

Please see results, line 154-158.

  • Both negative and positive control should be used (Fig. 3, 4, 5, 6 lacks positive controls to compare the effects of the azoles).

Response: We agree with the referee’s comment. We improved the figures 4,5, and 6 to address this issue.

(Figure 3): The negative and positive samples are not usually shown for cell cycle analysis for some reason. The cell cycle analysis requires DNA staining with propidium iodide that is stoichiometric, binding in proportion to the amount of DNA present in the cell. Then, the effects of any stimulus on the cell cycle distribution are commonly compared with the untreated control cells. There is no need to check how the cells appear when they show the cell cycle arrest in a specific phase.

Usually, the positive control samples are used by new users to become more familiar with the acquisition procedure of the flow cytometer, an asynchronously proliferating culture could be used as a positive control for flow cytometer set-up such as adjusting the sensitivity of photomultiplier tubes for propidium iodide staining such that the 2N and 4N peaks from singlet cells are centered at 200 and 400 (arbitrary units) on the X-axis.

  • 7 – some unification of the figures should be included (the figure for catalase looks different than other figures which introduces chaos).

Response: Depending on the raw data characteristics, the appropriate statistical analyses can converge to parametric and non-parametric methods. Therefore, the Sigma plot 11.0 software analyzes raw data for Normality and Equal Variance and then it proceeds for the best analysis. Therefore, ANOVA on ranks (Kruskal-Wallis test) has been used for CAT analyses and in this case Box Plot Graphs are the best way to represent data.

  • The Authors do not show any relationship between oxidative stress and the function of the cell model used, they do not investigate the effect of azole-induced oxidative stress on Sertoli cell function. Additional studies on Sertoli cells could be performed to evaluate if oxidative stress affects the Sertoli cell function.

Response: We thank the reviewer for this comment but his/her request to evaluate the relationship between oxidative stress and SCs function (e.g. to evaluate levels of Androgen Binding Protein and inhibin B, or BTB-associated elements and so on), at this time, goes beyond the purpose of the study. Our study is a preliminary work though to address specific questions such as the in vitro combinatorial cytotoxic effects of azoles on proliferation of Sertoli cells and intracellular mechanisms that mediate their chemical citotoxicity. On the other hand, Sertoli cells undergo apoptosis following azoles insult, thus suggesting the possibility of male subfertility/infertility to take place.

However, in the Introduction and Discussion sections we have added some sentences explaining the relationship between OS and Sertoli cell functions. In this regard, we have indirectly showed this connection. Indeed, in spermatogenic cells, Cys is used to synthesize protamine rather than GSH itself. Accordingly, the GSH content as well as SOD reservoir of spermatozoa are very low. GSH supplementation from Sertoli cells is required for spermatogenic cells both as protection from ROS and as an amino acid source for spermatogenesis. In this sense, an important function of Sertoli cells is to provide GSH by a direct interaction. The significant role of GSH from Sertoli cells in the supply of cysteine is supported by data on GGT(γ- glutamyl transpeptidase)-deficient mice. The GGT-knockout mouse has reduced testis and seminal vesicle sizes and is severely oligozoospermic and infertile. In line with these findings, under our experimental conditions, the presence of azoles induces a significant reduction of intracellular GSH levels, thus exhibiting a Sertoli cells dysfunction.

See revised discussion, lines 405-414.

  • Discussion section lacks the discussion supported with the published literature data. The results are not comprehensively interpreted considering other compounds present in the environment, their concentrations, the importance of oxidative status, and the Sertoli cell function.

Response: We agree with the referee’s comment. We improved the text to address this issue. Please see Introduction, lines 84-94, and discussion, lines 309-320.

  • Conclusions should be improved. I do not see any specific conclusions from the manuscript. Questions should be answered: Do environmental and biofluid azole concentrations affect the oxidative status in Sertoli cells? Which mechanisms are affected? How can we interpret the interactions between the azoles? Does oxidative stress affect Sertoli cell function?

Response: See revised conclusions, lines 415-423.  

  • As a general, English editing is needed. Commas in the figures should be converted to dots.

Response: We improved the text and figures to address this issue.

Round 2

Reviewer 1 Report

Figure 3B: The percentage of cells in each phase shown in Fig.3 A seems not like the authors shown in Fig. 3B. For example, in PCZ/KCZ group, the percentage of cells in G2 phase is 7%, in PCZ/MCZ group, the percentage of cells in G2 phase is also about 7%, whereas the G2 phase peak in PCZ/KCZ looks smaller than PCZ/MCZ. Please explain it.

Regarding the vehicle control issue, as the authors mentioned, you want to excluded the negative effect of DMSO, which means DMSO might cause some unpredictable effect, despite the its cytotoxic is not significant. Even 0.1% of DMSO induces alterations in microRNA and gene expression (Sci Rep. 2019 Mar 15;9(1):4641.). Therefore, it is not acceptable to use CTR to represent the absence of azoles instead of vehicle control.

The authors still not explain the purpose why you want to address the cytotoxic effects of different antifungal azoles combination? Are these combinations common in anti-fungal therapy?

Author Response

REVIEWER 1

  • Figure 3B: The percentage of cells in each phase shown in Fig.3 A seems not like the authors shown in Fig. 3B. For example, in PCZ/KCZ group, the percentage of cells in G2 phase is 7%, in PCZ/MCZ group, the percentage of cells in G2 phase is also about 7%, whereas the G2 phase peak in PCZ/KCZ looks smaller than PCZ/MCZ. Please explain it.

Response: The cell cycle profiles shown in figure 3B are generated by cell cycle analysis software (Modfit LT for Mac V3.0). They are representative of one of the independent experiments on the effect of PCZ/KCZ or PCZ/MCZ combinations and are not necessarily coincident with the mean value of the range but however within the indicated range in the pie chart. However, G2 for PCZ/KCZ is 7 ± 2.4, and G2 for PCZ/MCZ is 7 ± 3.1 and these values are not significant when compared with each other.

  • Regarding the vehicle control issue, as the authors mentioned, you want to excluded the negative effect of DMSO, which means DMSO might cause some unpredictable effect, despite the its cytotoxic is not significant. Even 0.1% of DMSO induces alterations in microRNA and gene expression (Sci Rep. 2019 Mar 15;9(1):4641.). Therefore, it is not acceptable to use CTR to represent the absence of azoles instead of vehicle control.

Response: In our study, vehicle CTR has been used as our internal control in preliminary experiments (e.g., cytotoxicity, cell cycle, mitochondrial activity and annexin-V assay). Accordingly, we re-evaluated the previously obtained data including vehicle CTR instead of CTR. For the other experimental conditions (e.g., enzymatic activities, GSH determination, ROS analyses, cPARP expression profiles) we added further experiments with vehicle CTR. Therefore, new statistical analyses were performed and new figures and graphs, substituting CTR with vehicle CTR (vCTR) have been inserted. In this regard, in our experimental conditions, we excluded any negative effects of DMSO that is biocompatible with no biological effect per se. We hope that we have definitely addressed this issue.

  • The authors still not explain the purpose why you want to address the cytotoxic effects of different antifungal azoles combination? Are these combinations common in anti-fungal therapy?

Response: We apologize for the misunderstanding.

Triazole and imidazole are worldwide produced in large quantities and extensively used in agricultural and industrial applications. In China they represent the second largest fungicide class by market value globally (WBISS Consulting Co, Ltd, 2016). Therefore, the agro-industrial compound PCZ is an environmental pollutant with high chemical and photochemical stability as well as low biodegradability, and its residue is frequently found in soil, agricultural products, water, and foods.

Moreover, azoles are used as active ingredients in pharmaceutical and personal care products. Thus, the pharmaceutical azoles FCZ, MCZ, and KCZ, are commonly used for the treatment of surface and deep mycoses but only individually or combined with others different antifungal drugs or non-antifungal agents (e.g., triazole with echinocandin, triazole with AmB, caspofungin and voriconazole) that may potentiate their antifungal activity (Scorzoni et al, 2017).  

Therefore, in our study, azole mixtures have been developed combining the agro-industrial compound (PCZ) vs the pharmaceutical azoles FCZ, MCZ, or KCZ. We could not explore the combined effects induced by MCZ, KCZ, and FCZ since these drug combinations are not used in clinical practice. We hope that we have clarified the misunderstanding.

Reviewer 2 Report

  1. The manuscript definitely has improved. The authors did a very good job addressing suggestions to the previous version of the manuscript.
  2. Authors’ Response: ‘Taking into consideration the cell density parameter (1 × 104 cells/cm2), Sertoli cell reached a confluence close to 80% at 48 h of incubation. Therefore, it is not possible to perform experiments at 72 and 96 h” I have also cultured TM4 cells and I agree that they grow very fast. However, it is possible - the Authors should only adjust the cel density to the duration of the exposure (here: seed less cells). 
  3. This research is about one particular cell line model. However, I do not see any reason to study cell line model without any reference to biological significance. As the results of the study are not validated on second cell line, they do not provide any information about Sertoli cells in general. In my opinion, this is the weakest point of the research.
  4. Figure 5 – positive control should be placed in the same graph as studied compounds, the conditions should be similar (why the incubations for studied compound was 30 min and 3 h and for the positive control was 1.5 h?), and the y axis should be also the same,
  5. Figure 7 –I am aware of using different statistical tests for different data. I only suggest to unify the graphs (add horizontal lines for catalase, adjust frames, replace commas with dots in numbers on axis).

Author Response

REVIEWER 2

  1. The manuscript definitely has improved. The authors did a very good job addressing suggestions to the previous version of the manuscript.

Response: We thank the reviewer.

  1. Authors’ Response: ‘Taking into consideration the cell density parameter (1 × 104 cells/cm2), Sertoli cell reached a confluence close to 80% at 48 h of incubation. Therefore, it is not possible to perform experiments at 72 and 96 h” I have also cultured TM4 cells and I agree that they grow very fast. However, it is possible - the Authors should only adjust the cel density to the duration of the exposure (here: seed less cells). 

Response: In our opinion, it is important do not change the initial cell density (recommended by ECACC cell line Data Sheet) since deviations in this parameter may affect cell functions, resulting in different experimental results and thus, to be a source of variability (Trajkovic et al., 2019). Cellular decisions often depend on threshold concentrations; feedback mechanisms can generate sharp transitions in cellular states that are initiated by small concentration changes of critical regulators. In this sense, mounting evidence has been reported showing the influence of cell density on cellular biochemistry and physiology (Trajkovic et al., 2019).

  1. This research is about one particular cell line model. However, I do not see any reason to study cell line model without any reference to biological significance. As the results of the study are not validated on second cell line, they do not provide any information about Sertoli cells in general. In my opinion, this is the weakest point of the research.

Response: we thank the reviewer for his/her comment. In further studies we will carefully evaluate this suggestion. 

  1. Figure 5 – positive control should be placed in the same graph as studied compounds, the conditions should be similar (why the incubations for studied compound was 30 min and 3 h and for the positive control was 1.5 h?), and the y axis should be also the same,

Response: As suggested, we uniformed y axis and placed positive control in each graph.

The positive control is used only to ensure diagnostic accuracy and is determined experimentally for each cell line being tested. For TM4 cells, in our experimental conditions, treatment with 500 µM t-BHP (for 1.5 h) represents the time and the optimal final concentration of positive control. Therefore, the positive control was performed concurrently with the other experimental samples, each time the in vitro method was performed. However, the times were synchronized to allow simultaneous reading of the samples.

  1. Figure 7 –I am aware of using different statistical tests for different data. I only suggest to unify the graphs (add horizontal lines for catalase, adjust frames, replace commas with dots in numbers on axis).

Response: As suggested we modified Figure 7. Please, take into account that we have had to follow reviewer 1 suggestions. Thus, we have added some experiments that include vehicle (vCTR), also leading to  new statistical analyses. However, results structure did not change.  

Round 3

Reviewer 1 Report

No.